# CRISPR Deletion of a SVA Retrotransposon Demonstrates Function as a *cis*-Regulatory Element at the *TRPV1/TRPV3* Intergenic Region

**DOI:** 10.3390/ijms22041911

**Published:** 2021-02-15

**Authors:** Emma Price, Olympia Gianfrancesco, Patrick T. Harrison, Bernhard Frank, Vivien J. Bubb, John P. Quinn

**Affiliations:** 1Department of Pharmacology and Therapeutics, Institute of Systems, Molecular & Integrative Biology, University of Liverpool, Liverpool L69 3GE, UK; emma.price2@nih.gov (E.P.); olympia.gianfrancesco@igmm.ed.ac.uk (O.G.); jillbubb@liverpool.ac.uk (V.J.B.); 2MRC Human Genetics Unit, Institute of Genetics and Molecular Medicine, University of Edinburgh, Edinburgh EH4 2XU, UK; 3Department of Physiology, BioSciences Institute, University College Cork, Cork, Ireland; p.harrison@ucc.ie; 4Department of Pain Medicine, Walton Centre NHS Foundation Trust, Liverpool L9 7LJ, UK; bernhard.Frank@liverpool.ac.uk

**Keywords:** retrotransposon, SVA, *cis*-regulatory element, *TRPV1*, *TRPV3*, CRISPR, gene expression

## Abstract

SINE-VNTR-*Alu* (SVA) retrotransposons are a subclass of transposable elements (TEs) that exist only in primate genomes. TE insertions can be co-opted as *cis*-regulatory elements (CREs); however, the regulatory potential of SVAs has predominantly been demonstrated using bioinformatic approaches and reporter gene assays. The objective of this study was to demonstrate SVA *cis*-regulatory activity by CRISPR (clustered regularly interspaced short palindromic repeats) deletion and subsequent measurement of direct effects on local gene expression. We identified a region on chromosome 17 that was enriched with human-specific SVAs. Comparative gene expression analysis at this region revealed co-expression of *TRPV1* and *TRPV3* in multiple human tissues, which was not observed in mouse, highlighting key regulatory differences between the two species. Furthermore, the intergenic region between *TRPV1* and *TRPV3* coding sequences contained a human specific SVA insertion located upstream of the *TRPV3* promoter and downstream of the 3′ end of *TRPV1*, highlighting this SVA as a candidate to study its potential *cis*-regulatory activity on both genes. Firstly, we generated SVA reporter gene constructs and demonstrated their transcriptional regulatory activity in HEK293 cells. We then devised a dual-targeting CRISPR strategy to facilitate the deletion of this entire SVA sequence and generated edited HEK293 clonal cell lines containing homozygous and heterozygous SVA deletions. In edited homozygous ∆SVA clones, we observed a significant decrease in both *TRPV1* and *TRPV3* mRNA expression, compared to unedited HEK293. In addition, we also observed an increase in the variability of mRNA expression levels in heterozygous ∆SVA clones. Overall, in edited HEK293 with SVA deletions, we observed a disruption to the co-expression of *TRPV1* and *TRPV3*. Here we provide an example of a human specific SVA with *cis*-regulatory activity in situ, supporting the role of SVA retrotransposons as contributors to species-specific gene expression.

## 1. Introduction

SINE-VNTR-*Alus* (SVAs) are the evolutionarily youngest family of transposable elements (TEs) currently characterized within the human genome. The SVA family emerged throughout primate evolution (Figure 1A) and belong to a group of TEs termed non-long terminal repeat (LTR) retrotransposons (which also includes long interspersed nuclear element 1 (LINE-1) and *Alu*), which collectively remain the only actively mobile TEs in the human genome [1,2]. Mobilization of non-LTR TEs leads to novel insertions which contribute to genome sequence and structure—ultimately contributing to genome evolution [3,4,5]. The insertion site of TEs can affect gene regulation through multiple mechanisms including the introduction of transcription factor binding sites (TFBSs), novel transcriptional start sites (TSS), alternative splicing, exonization, and alterations to epigenetic marks including DNA methylation and histone modifications [6,7,8,9,10]. Thus, novel TE insertions can be co-opted for new regulatory functions which drives species-specific gene expression. In recent years, attention has turned towards the regulatory impact of SVAs in the human genome. Previously, we have demonstrated the transcriptional regulatory properties of isolated SVA sequences utilizing reporter gene assays [11,12]. More recently, bioinformatic studies utilizing ChIP-seq and RNA-seq datasets generated from liver across different primate species have identified that newly evolved (i.e., species-specific) *cis*-regulatory elements (CREs) were enriched in SVA sequences, highlighting SVAs as potentially important contributors to gene regulation in primates [13,14]. Candidate SVA CREs were verified using reporter gene assays, which lent support to the hypothesis that SVAs can have regulatory properties but this did not necessarily confirm regulatory function in situ in the human genome, and more specifically in the context of endogenous genes [13,14].

A direct approach to address the role of candidate regulatory domains in situ can be performed through CRISPR (clustered regularly interspaced short palindromic repeats)/Cas9 (CRISPR associated protein 9)-mediated deletion of entire TE sequences. An exemplar of this approach was seen in the investigation of the etiology of neurodegenerative disorder X-linked dystonia parkinsonism (XDP). In this study, an XDP-specific SVA insertion was shown to cause intron retention and reduced expression of the gene *TAF1*, which was causative of the disorder [15,16]. Upon CRISPR/Cas9 deletion of the SVA in patient derived induced pluripotent stem cells (iPSCs), aberrant splicing and *TAF1* expression were rescued, implicating the SVA in disease-associated gene regulation. Herein, our specific aim was to extend this approach and demonstrate *cis*-regulatory effects of an SVA in a non-disease context, by highlighting the contribution of an endogenous SVA to human specific gene expression.

We and others have previously demonstrated that SVAs preferentially insert into gene dense regions and several regions across the human genome are enriched for SVA insertions [17]. One such region previously identified was a 1 Mb region (chr17:3000001–4000000, hg19) on chromosome 17 (chr17p13.2), which contained six SVA insertions (Figure 1B). These insertions were SVA subclass D, which is a subclass that shares some insertions with gorilla and chimpanzee but the majority (67.5%) are human specific [3]. Furthermore, SVA D comprises 44.4% of all SVAs in the human genome [17], highlighting this subclass as particularly active throughout human evolution. It has been established that there is a strong bias of human specific SVAs inserting into regions already containing SVAs [11,12,17]. All SVAs at chr17p13.2 were subclass SVA D (Figure 1C), with 5 out of 6 being full length and human-specific (Figure 1B), highlighting chr17p13.2 as an area of the genome that has been tolerant to multiple SVA insertions throughout recent human evolution and an area of the genome where human-specific gene regulation via SVA-mediated activity may have evolved. Given the enrichment of human-specific SVAs at this region, in such close proximity to many genes, it provided a region of the genome with multiple potential human specific CREs to explore further.

To identify a candidate SVA at this location with the potential to function as a *cis*-regulatory element, we explored the location of SVAs at chr17p13.2 with respect to their adjacent genes and at the same time, compared RNA-seq data from humans and mice. This approach identified an SVA insertion, proximal to genes *TRPV1* and *TRPV3*, which displayed human-specific expression patterns. *TRPV1* and *TRPV3* encode polymodal transient receptor potential channels which enable thermosensory perception and have well established roles in pain and inflammation [18,19]. Our preliminary data obtained from reporter gene assays suggested this SVA was indeed regulatory, and thus highlighted this SVA as a candidate for further study. To determine *cis*-regulatory activity in situ, we devised a dual-targeting CRISPR/Cas9 strategy to delete the SVA in the HEK293 cell line and measured effects on *TRPV1* and *TRPV3* gene expression. In this study, we deleted the entire SVA sequence in multiple HEK293 clones and demonstrated gene expression changes in comparison to unedited cells, providing in situ functional data that supported the role of SVAs as CREs in the human genome and highlighted their role in the evolution of gene regulation in humans.

## 2. Results

### 2.1. Intergenic Region between TRPV1 and TRPV3 Contains a Human Specific SVA Insertion Predicted to Function as a Regulatory Domain

In the preliminary bioinformatic analysis of SVA insertions across the chr17p13.2 region (Figure 1C), the SVA at the *TRPV1* and *TRPV3* locus was notable as it was located within the intergenic region containing the *TRPV3* promoter and directly adjacent to other prominent histone marks indicative of regulatory domains (Figure 1E). A more detailed overview of other SVA insertions in respect to other genes is provided in Appendix A. Analysis showed that the SVA and both *TRPV1* and *TRPV3* coding sequences, were all encoded on the antisense strand of DNA, with the SVA located approximately 400 bp downstream of the *TRPV1* 3′UTR and 5.7 kb upstream of the 5′ *TRPV3* TSS (Figure 2A). The intergenic region contained numerous repetitive DNA sequences that were conserved with other primate species (Figure 2A,B). In total, 73% of the 7.4 kb intergenic sequence was comprised of TEs including multiple SINEs (specifically *Alus*) (36%) and LINEs (12%) (Figure 2B). The single SVA insertion (chr17:3466973–3468374, hg19) was the largest TE (1402 bp) in this region, accounting for a substantial proportion (19%) of the intergenic sequence (Figure 2B). In addition, the SVA was directly adjacent to an evolutionary conserved region (ECR) (chr17:3466258–3466820 hg19) containing a mammalian-wide interspersed repeat (MIR) (Figure 2A). MIRs are the most ancient family of TEs in the human genome, enriched for TFBSs and have been shown to function as enhancers in vitro [20]. ENCODE histone data overlaid across this ECR was indicative of regulatory activity and it was also listed as a candidate CRE of *TRPV1* and *TRPV3* within ENCODE (ENCODE acc. no. EH38E1841572) (Figure 2A). No histone data were available for the SVA itself because larger repetitive sequences are difficult to map in short read sequence data, thus they are often excluded from such analyses (Figure 2A). Nevertheless, the available bioinformatic data supported that the ECR was a functional CRE and we reasoned that its modulation in humans may be impacted upon by the adjacent human specific SVA insertion.

To address human-specific gene expression, we compared the expression of all protein coding genes encoded at chr17p13.2 (Figure 1C) with orthologous genes encoded at the syntenic region (33.4% of bases and 99.4% of span) chr11(qB4–qB5) in the mouse genome (Figure 1D). Utilizing data from an RNA-seq study which compared gene expression across multiple tissues between human and mouse [21], expression differences in various genes at chr17p13.2 were noted (e.g., *ITGAE*, *P2RX1*, *P2RX1*) (Figure 2C). However, the regulation of *TRPV1* and *TRPV3* was our focus, due to the organization of the surrounding regulatory domains with respect to the SVA insertion (Figure 2A). We observed widespread expression of *TRPV1* and *TRPV3* across multiple human tissues in comparison to a tissue restricted expression profile in the mouse (Figure 2C). We observed *hTRPV1* expression was generally at relatively low levels but with high expression in dorsal root ganglia (DRG). Similarly, *mTrpv1* was also expressed at relatively high levels in DRG, however *mTRPV1* expression was in general more restricted, with low levels of expression only documented in nucleus accumbens and skeletal muscle. *hTRPV3* expression was seen in most tissues analyzed however *mTrpv3* was restricted to spinal cord. This analysis also highlighted the co-expression of *TRPV1* and *TRPV3* in many human tissues compared to mice, suggesting differences in key regulatory mechanisms at the molecular level. We hypothesized that the SVA may be functional as a *cis*-regulatory domain contributing to the regulatory differences of *TRPV1* and *TRPV3* observed in humans. 

### 2.2. Reporter Gene Assays Support Regulatory Potential of SVA at TRPV1/TRPV3 Locus

To confirm the regulatory potential of the SVA and the adjacent ECR, both sequences were cloned into the reporter gene construct pGL3-P, upstream of the minimal SV40 promoter in both forward and reverse orientations—with forward resembling the endogenous orientation of the SVA with respect to *TRPV1* and *TRPV3* (Figure 2D). The sequences and coordinates of the cloned SVA and ECR fragments are given in Appendix A. Luciferase activity was measured 48 h post-transfection. In HEK293 transfected with the SVA reporter gene constructs, a statistically significant 2-fold decrease (*p* < 0.05) in luciferase activity was observed when the SVA was cloned in the forward orientation. Similarly, a smaller yet still significant 1.6-fold decrease (*p* < 0.05) when cloned in the reverse orientation when compared to the unmodified pGL3-P vector was observed, indicating the SVA was functional as a transcriptional regulatory domain in this cell line, independent of orientation (Figure 2D). The ECR reporter gene construct displayed a small 1.3-fold increase (*p* < 0.05) in luciferase activity when the ECR was cloned in the forward orientation but no difference was observed when the ECR was cloned in the reverse orientation when compared to unmodified pGL3-P (Figure 2D).

### 2.3. Dual-Targeted CRISPR/Cas9 Deletion of the SVA in HEK293 

To address the hypothesis in the context of the endogenous genes, expression of *TRPV1* and *TRPV3* was confirmed in the HEK293 cell line (Appendix A). HEK293 was chosen as a model cell line to conduct CRISPR due to its high transfection efficiency, which was found to be a limiting factor in the genome editing efficiency in other cell lines we tested (e.g., HAP1 and SH-SY5Y) during early stages of protocol development (data not shown). Upon confirmation both genes were active in this cell line, a dual-targeted CRISPR strategy to delete the entire SVA sequence in HEK293 cells was developed to subsequently measure the potential impact on *TRPV1* and *TRPV3* expression (Figure 3A) [22]. Two guide RNAs (gRNAs) were designed, which targeted sequences 66 bp downstream and 213 bp upstream of the SVA, and cloned into the Cas9 expression vector pSpCas9(BB)-2A-GFP, resulting in two separate vectors each containing a single gRNA sequence (Figure 3A) [23]. The dual-target strategy, based on co-transfection of the two independent Cas9 vectors each containing a specific gRNA tag, was predicted to result in generation of two double strand breaks (DSBs) at positions chr17:3563606 and chr17:3565282 (hg19), following which the intermediate sequence containing the SVA would be lost and the ends repaired via non-homologous end joining (NHEJ). This approach was designed to facilitate the deletion of a 1677 bp sequence containing the SVA (1402 bp) (Figure 3A).

Edited HEK293 cells were expanded as clonal cells lines and CRISPR-edited genetic regions were screened via PCR to identify clones containing the desired SVA deletions (Figure 3B). PCR products for unedited (containing SVA) and edited (deleted SVA) regions were 2486 bp and 808 bp in length, respectively (Figure 3B). Following transfection, 215 clonal cell lines were screened in total. Three independent clonal cell lines (<2%) amplified only edited PCR products, indicating all SVA alleles were successfully deleted, and were termed homozygous ∆SVA clones. In addition to the desired homozygous ∆SVA HEK293 genotype, several clonal cell lines (10%) amplified both unedited and edited PCR products, indicating that these clones carried an intact SVA and an SVA deletion (Figure 3B). These clones were termed heterozygous ∆SVA clones and three (chosen at random) were included in subsequent analysis. As an additional control, HEK293 clones transfected with non-target gRNAs (ntgRNAs) were also generated; these guides were specifically designed to not recognize any human DNA sequence, thus should not guide the Cas9 to any specific sequence or result in any modifications.

Sequencing of the PCR products across the predicted DSB breakpoints was performed in all homozygous and heterozygous ∆SVA clones. In all three homozygous ∆SVA clones, the breakpoints were identified at the predicted DSB sites with no indels, highlighting effective and accurate repair at the edited sites (Figure 3C). This data confirmed the predicted modifications had been generated in these clones, with a precise deletion of 1677 bp containing the SVA. Sequencing of the heterozygous ∆SVA clones demonstrated that only clone 3 had breakpoints at the predicted DSB sites. Clone 1 and 2 carried slightly larger deletions (Figure 3C). Breakpoints in clone 2 extended 2 bp and 16 bp beyond the predicted gRNA1 and gRNA2 DSB sites, respectively. Clone 1 carried the largest deletion, with breakpoints extending 4 bp and 16 bp beyond the predicted gRNA1 and gRNA2 DSB sites, respectively. These data confirm that the desired modifications were successfully generated in homozygous ∆SVA clones and additional modifications had occurred in heterozygous ∆SVA clones which were also taken forward for gene expression analysis.

### 2.4. TRPV1 and TRPV3 Expression Was Disrupted in CRISPR Edited HEK293 Clones Containing SVA Deletions

To assess if the SVA was functional as a CRE, total mRNA expression levels of adjacent genes *TRPV1* and *TRPV3* were measured using qPCR (Figure 4). Validation of qPCR products and primer efficiencies are shown in Appendix A. Expression levels in edited clonal cell lines were compared against unedited HEK293. All relative expression levels were normalized against reference gene *ACTB*. Relative expression values were plotted as log2 fold change to enable comparative visualization of increases and decreases in expression levels. HEK293 cells transfected with ntgRNAs showed negligible difference in *TRPV3* levels (*p* > 0.05), when compared to unedited cells, indicating that any changes observed in edited cells could be directly attributed to deletion of the SVA and not due to transfection of CRISPR machinery without genome modification (Figure 4A). We did observe a slight decrease in *TRPV1* in ntgRNA cells, however this was negligible and not determined to be statistically significant (*p* > 0.05) (Figure 4A). In homozygous ΔSVA clones, a significant decrease in *TRPV3* mRNA expression (*p* < 0.05) was observed (Figure 4B). A decrease in *TRPV1* expression was also seen that was determined to be statistically significant (*p* < 0.05), however it was relatively small compared to unedited cells and also similar to the small decrease in cells transfected with ntgRNAs. This implied that, under the experimental conditions employed, *TRPV1* expression between individual HEK293 clones was minimally affected by the presence or absence of the SVA. However, a consistent and much greater decrease in *TRPV3* expression was observed. These results provided evidence which supported the role of this SVA as a CRE at the intergenic region between *TRPV1* and *TRPV3* in HEK293.

In contrast with homozygous ΔSVA HEK293 clones, which had shown relatively consistent decreases in *TRPV1* and *TRPV3* mRNA levels, highly variable gene expression results between individual heterozygous ΔSVA HEK293 clones were observed (Figure 4B). For example, *TRPV1* expression did not change in clone 2, whereas clone 1 and clone 3 both showed a decrease in *TRPV1* expression. When we measured *TRPV3* levels, clone 1 showed a large increase, clone 2 showed a small increase, and clone 3 showed a small decrease (*p* > 0.05). No statistical significance was determined for either *TRPV1* or *TRPV3* expression, however the overall trend observed in heterozygous ΔSVA clones was an increase in mRNA expression variability in both genes.

Analysis of RNA-seq data had shown co-expression of *TRPV1* and *TRPV3* in multiple human tissues (Figure 2C), therefore the ratio of *TRPV1:TRPV3* in unedited and edited cell lines was examined (Figure 4C). A weak positive correlation between *TRPV1* and *TRPV3* expression in unedited HEK293 cells was identified and a strong positive correlation in cells transfected with ntgRNAs (Figure 4D). It should be noted that a small decrease in *TRPV3* expression in one unedited replicate was observed, however *TRPV1* expression remain consistent across all unedited replicates. Given the small range in expression values between replicates, this small decrease in *TRPV3* was enough to decrease the strength of the positive correlation in unedited replicates. However, this trend of positive correlation in unedited cells was lost in all edited clones. Homozygous ΔSVA clones showed a weak negative correlation between *TRPV1* and *TRPV3* expression and no correlation was observed in heterozygous ΔSVA clones (Figure 4D). Across individual homozygous ΔSVA clones consistently increased levels of *TRPV1* compared to *TRPV3* were observed, which was not seen in unedited cells, indicating that co-expression of *TRPV1* and *TRPV3* was disrupted in edited cells completely lacking the SVA insertion (*p*-value > 0.05) (Figure 4C). Due to the variability of *TRPV1* and *TRPV3* expression in heterozygous ΔSVA clones, no clear directional change in ratio was evident however the overall trend was an increase in variability (*p*-value > 0.05). It should be acknowledged that the statistical power in this study was limited due to the number of biological replicates (*n* = 3). These data further support that the regulatory mechanism contributing to the co-expression of *TRPV1* and *TRPV3* observed in unedited cells, was not maintained in the absence of the SVA.

## 3. Discussion

### 3.1. Human Specific SVA Insertion at the TRPV1/TRPV3 Locus Identified as a Candidate CRE

In this study, we identified an SVA (subclass D) at the *TRPV1/TRPV3* intergenic region, which we hypothesized was functional as a human specific CRE following comparative gene expression analysis showing species differences in *TRPV1* and *TRPV3* regulation between human and mouse. The differential expression of *TPRV3* between human and mouse has been previously reported in the literature, with a focus on roles in the nervous system (e.g., expression in DRG) [19,24], however the data analysis conducted here suggested a broader physiological role for both TRPV1 and TRPV3 in many more human tissues. The results from our reporter gene assays conducted in HEK293 showed a repressive effect of the SVA when cloned into the pGL3-P system. These findings were consistent with data previously published by our group, showing repressive effects of other SVAs cloned into the same pGL3P system when tested in clonal cell lines SH-SY5Y and SKNAS [11,12]. Furthermore, our data is also consistent with that published by Trizzino et al., who cloned SVAs into the pGL4.23 system and conducted reporter assays in HepG2 cells [13,14]. Furthermore, a study of repression in exogenous SVA reporter constructs showed repressive binding of the human silencing hub (HUSH) complex to the central VNTR, which induced methylation at the SV40 promoter [25]. Overall, trends from various reporter models show a repressive effect of SVAs. However, this does not necessarily reflect the endogenous role of SVAs in situ, as they are simplistic and cannot account for additional factors like chromatin structure or adjacent regulatory sequences. Interestingly, the ECR adjacent to the SVA displayed all the hallmarks of a regulatory domain, therefore we expected to see greater changes in luciferase than those observed in this experiment. Nevertheless, when comparing results from both the ECR and SVA in HEK293, the SVA exerted a stronger transcriptional effect, leading us to further investigate its potential function as a CRE using CRISPR/Cas9. 

### 3.2. CRISPR-Cas9 Deletion of SVA in HEK293 Generated Homozygous and Heterozygous Clones

Using a dual-target CRISPR/Cas9 approach, we successfully generated edited HEK293 clones with deletion of all SVA alleles (homozygous ΔSVA) and clones demonstrating the presence and absence of SVA alleles (heterozygous ΔSVA). Attempts were made to also delete the ECR using CRISPR to assess for endogenous effects that were not apparent in the reporter gene assays, however this was unsuccessful. A limitation of this method was the low yield of clones with the desired genetic modifications. We determined an editing efficiency of 10% for heterozygous ΔSVA clones and <2% for homozygous ΔSVA. The previously published *TAF1* SVA study reported a modification efficiency of 30%, but this was performed in patient derived cell lines which possessed only a single pathogenic SVA insertion polymorphism at the *TAF1* gene [16], in comparison to the SVA at *TRPV1/TRPV3* which is not an insertion-based polymorphic variant and thus required deletion of more than one copy. The lower modification efficiencies for homozygous ΔSVA clones, highlighted the difficult nature of deleting both copies of large DNA sequences. However, sequencing across edited regions in homozygous ΔSVA clones did reveal a high level of modification accuracy using this dual-target CRISPR/Cas9 system. 

### 3.3. TRPV3 Expression Was Significantly Decreased in Homozygous ΔSVA Clones

Following the deletion of all SVA alleles in homozygous ΔSVA clones, a significant decrease in *TRPV3* mRNA expression was seen compared to unedited cells, indicating that the presence of the SVA is contributing to the expression of *TRPV3* in HEK293. Whilst a significant decrease in *TRPV1* was also observed in homozygous ΔSVA clones compared to unedited cells, the level of decrease was similar to HEK293 transfected with ntgRNA controls, therefore the effects of the SVA deletion on *TRPV1* mRNA expression could not be determined. Nevertheless, co-expression of *TRPV1* and *TRPV3* is found in many human tissues, which is not true of mouse tissues (e.g., DRG) which have been shown to express *TRPV1* but not *TRPV3* (Figure 2C). These functional data lend support to the hypothesis that SVAs serve as newly evolved CREs in primate genomes and contribute to gene regulation in primate species [13,14,17,26]. When we compared the ratio of *TRPV1:TRPV3* expression between unedited and homozygous ΔSVA clones, a greater decrease in *TRPV3* expression compared to *TRPV1* was observed. It should be noted however that the greater effect observed on *TRPV3* should be considered within this model only, and that differential regulation of *TRPV1* and *TRPV3* may be possible in a tissue-specific and stimulus-inducible fashion. In this model however, it is interesting that closer proximity of the SVA was much closer to the promoter of *TRPV3* (~5 kb) than to that of *TRPV1* (~27 kb). A recent study supporting the role of SVAs as proximal CREs showed that silencing of SVAs via induction of H3K9me3 marks, resulted in greater deregulation of genes with TSSs in close proximity (0–5 kb) to the SVA, than compared to genes with TSSs situated farther away (>100 kb) [26]. An alternative explanation is that the SVA is in even closer proximity (400 bp) to the 3′UTR of *TRPV1*. The 5′ end of the SVA contains a CT rich hexamer domain which contain MAZ-binding sites, which can affect polyadenylation signals and gene regulation—also in a tissue dependent manner [27]. Therefore, it would be interesting to quantify TRPV1 and TRPV3 protein levels to determine if there is any effect on mRNA stability and subsequent protein levels. 

### 3.4. TRPV1 and TRPV3 mRNA Expression Was Highly Variable in Heterozygous ΔSVA Clones

Gene expression in heterozygous ΔSVA clones was highly variable, unlike homozygous ΔSVA clones that demonstrated consistent decreases in expression of *TRPV1* and *TRPV3*. No statistical significance was determined for either *TRPV1* or *TRPV3* in heterozygous ΔSVA clones, however the overall trend in heterozygous ΔSVA clones was an increase in mRNA expression variability of both genes. Furthermore, unlike homozygous ΔSVA clones, there was no correlation between gene expression values in heterozygous Δ SVA clones (Figure 4F). This indicated that loss of an SVA at the intergenic region between both genes may be disruptive to the regulatory mechanisms that drive co-expression in unedited cells. Deletion breakpoints were consistent in all homozygous ΔSVA clones but variable in heterozygous ΔSVA clones. It is hypothesized that this may have contributed to the observed differences. Consistent with this hypothesis, heterozygous ΔSVA clone 3 carried the same breakpoints in the edited region as all homozygous ΔSVA clones and showed expression of *TRPV1* and *TRPV3* that was consistent with the homozygous ΔSVA clones. However, heterozygous clones 1 and 2 showed increases specifically in *TRPV3* expression. Interestingly, heterozygous clones 1 and 2 carried breakpoints in the edited region that resulted in a slightly larger deletion, which extended 10 bp into a (CA)_n_ repeat. This suggested that there was potentially unresolved regulatory potential in this additional sequence. Consideration must also be paid to the remaining SVA alleles in heterozygous ΔSVA clones. PCR genotyping indicated that an SVA allele was still present, however this did not guarantee that it was unedited. For example, DSBs could have been created and then repaired, failing to result in excision of the SVA, however indels or inversions could have occurred. We sequenced across breakpoints of the remaining SVA alleles following the CRISPR modification process. Annotated DNA sequences are given in Appendix A. We found SNPs and a lack of alignment in clone 1 and 2 with the reference genome (Appendix A), indicating introduction and repair of DSBs resulting in the SVA-containing sequence being retained, however some additional modifications had also taken place. The effect of these modifications is unclear; nevertheless, whilst not statistically significant, a clear trend emerged in heterozygous ΔSVA clones showing deregulation in both *TRPV1* and *TRPV3* expression when compared to unedited cells and the ratio of *TRPV1:TRPV3* was disrupted in heterozygous ΔSVA clones, consistent with homozygous ΔSVA clones. 

### 3.5. CRISPR Deletion of SVA at TRPV1/TRPV3 Locus Demonstrates in Situ Function as Newly Evolved CRE 

Overall, when taking all the data into account, deregulation of expression of *TRPV1* and *TRPV3* was observed in edited cells, regardless of homozygous or heterozygous ΔSVA deletions. The impact on gene expression following deletion of the SVA, supports its role as a CRE at the intergenic region between *TRPV1* and *TRPV3*. The mechanism by which an SVA was previously identified as regulatory in *TAF1* was intron retention which resulted in a decrease in *TAF1* expression [16]. This specific disease mechanism is a different context to that being explored here and whilst relevant, should be considered independently [15,16]. The exact mechanism by which the SVA at *TRPV1*/*TRPV3* functions remains to be determined however previous studies point to the recruitment of transcription factors [9,28]. Aberrant expression of *TRPV1* is implicated in multiple pain associated conditions including diabetic neuropathy [29], irritable bowel syndrome [30], chronic pancreatitis [31], vulvodynia [32], and *TRPV3* is elevated in inflammatory skin-related conditions like psoriasis [33,34]. Thus, *TRPV1* and *TRPV3* remain key targets for the development of pharmaceuticals [35]. However, there have been difficulties translating advances identified in preclinical studies utilizing mouse models [36]. To our knowledge, to date, no regulatory domain contributing to the human specific expression of *TRPV3* has yet been identified [37,38]. There are reports of TRPV1 and TRPV3 forming heteromeric channels in humans which are hypothesized to contribute to the fine tuning of sensory inputs, therefore the influence of the SVA and its role in *TRPV1/TRPV3* regulation may have contributed to this molecular phenotype in human tissues and contributed to difficulties in developing drugs that are translatable based on mouse models [39,40]. It must be noted that the intergenic region contained many other transposable elements (e.g., *Alus*) that may also be predicted to contribute to gene regulation in human and non-human primates [19,24], therefore the SVA would be a contributor in part to the full regulatory network of *TRPV1* and *TRPV3* observed in humans. However, it remained our focus to study the impact of a fairly recent SVA insertion—in terms of human genome evolution. In conclusion, the work presented here is the first reported example of a non-disease SVA being deleted using CRISPR and functional data supporting its role as a *cis*-regulatory domain that directly impacts mRNA expression of adjacent genes in vitro. To our knowledge, this is the first description of a human specific regulatory domain identified at the *TRPV1* and *TRPV3* locus with the potential to contribute to the human specific expression of *TRPV3* previously reported in the literature. These data give support to the role of SVAs as drivers of gene regulation and phenotypic evolution in primates, and shed light on the regulatory differences already identified between mice and humans at the *TRPV1* and *TRPV3* locus. 

## 4. Materials and Methods

### 4.1. Bioinformatic Analysis

A list of SVA enriched regions was obtained from supplementary data published by Gianfrancesco et al. 2019 [4]. A list of all protein coding genes at the SVA enriched region at chr17p13.2–3, specifically at coordinates chr17:3000001–4000000 (hg19) were obtained from UCSC genome browser. The same coordinates were used to view the syntenic gene region in the mouse genome at coordinates chr11:72843250–74363624 (mm10). A list of protein coding genes was exported from UCSC from both human and mouse genomes at these coordinates. The list of protein coding genes was filtered for 1:1 orthologues. Expression values for 1:1 orthologues across multiple tissues in humans and mice was obtained from the RNA-seq dataset published by Ray et al. 2018 (given as transcripts per million; TPM) [21]. Data for comparable tissues across human and mouse were filtered out and used for cross species comparison. Heatmaps reflecting relative TPM values across species were generated using Morpheus software, available at https://software.broadinstitute.org/morpheus/ (accessed on 11 November 2020).

### 4.2. Cell Culture

Both luciferase reporter assays and CRISPR modifications were conducted in HEK293 cells (ATCC CRL-1573). Cells were cultured in Dulbecco’s Modified Eagle Medium containing 4.5 g/L d-glucose and l-glutamine (Gibco, Paisley, UK) supplemented with; 10% FBS (Gibco), 1% 100 mM sodium pyruvate (Sigma, UK) and 1% penicillin/streptomycin solution (10,000 units penicillin and 10 mg streptomycin/mL) (Sigma). Cells were maintained in a humidified incubator at 37 °C with 5% CO_2_. 

### 4.3. Generating SVA and ECR Reporter Gene Constructs

The SVA and ECR sequences were amplified from HEK293 gDNA using PCR. PCR reactions consisted of the following reagents; 1× Green GoTaq reaction buffer (Promega, Southampton, UK), 2.5 mM MgCl2 (Promega), 0.2 mM dNTPs (Sigma), primers 0.1 mM (Sigma), 0.5 units GoTaq Hot Start Polymerase (Promega), gDNA template (10 ng), made up to a final volume of 20 µL with UltraPure™ DNase/RNase Free Distilled Water (Thermo Fisher Scientific, Gloucester, UK). Thermal cycles were performed using the SimpliAmp™ Thermal Cycler (Applied Biosystems, UK). Specific primers pairs and cycling conditions are listed in Appendix A. PCR products were visualized using 1–2% agarose gels stained with ethidium bromide solution (Sigma). Gels were visualized under UV light using the BioDoc-It Imaging System (UVP). Amplicons and restriction digest products were purified using the Wizard^®^ SV Gel and PCR Clean-Up System (Promega). Amplicons were initially ligated into the Dual promoter pCRII vector using the TA Cloning Kit (Invitrogen, Paisley, UK). The amplicon was then subcloned into the SacI and NheI sites of the reporter gene vector pGL3-Promoter (pGL3-P) (Promega), which encodes Firefly luciferase (FLuc). Ligations were completed using T4 DNA ligase (NEB). All cloning was propagated using Subcloning efficiency DH5α competent cells (Invitrogen), grown in LB medium supplemented with 100 μg/mL ampicillin. Plasmid DNA used for transfection was purified using the Plasmid Maxi Kit (QIAGEN, Manchester, UK). 

### 4.4. Luciferase Reporter Gene Assays

To evaluate the potential regulatory transcriptional activity of the SVA and ECR sequences, HEK293 cells were co-transfected with the relevant reporter gene plasmids and pRL-TK which encodes Renilla luciferase (RLuc) to enable normalization for transfection efficiency, using TurboFect Transfection Reagent (Thermo Fisher Scientific). pGL3-Basic (pGL3-B), which contains no promoter and should not express luciferase, was used as a negative control in transfection experiments. Transfections were repeated in four separate biological controls. Media was replaced 4 h post transfection and cells were assayed 48 h post transfection. FLuc and RLuc activity was measured in cell lysates using the Dual-Glo Luciferase Assay System (Promega). Relative light units (RLU) were detected using the GloMax 96 Microplate Luminometer. RLUs for each transfected culture were normalized against negative controls. The adjusted ratio of FLuc/RLuc was calculated for each condition and expressed as normalized Firefly luciferase activity (averaged across four repeats).

### 4.5. CRISPR/Cas9 Nuclease-Mediated Genome Editing

sgRNA sequences flanking the 5′ and 3′ end of the SVA were identified using http://crispr.mit.edu/ (accessed on 17 May 2018) based on the *S. pyogenes* Cas9 5′-NGG-3′ PAM recognition sequence. Target sequences are given in Appendix A. Suitable oligonucleotides 20 bases in length were modified by removing the PAM sequence at the 3′ end of the sense oligonucleotide, followed by addition of CACC at the 5′ end of the sense oligonucleotide, and addition of AAAC at the 5′ end of the antisense oligonucleotide to generate *BbsI* overhangs. Complimentary oligonucleotides were annealed together to create a double stranded insert which was then ligated into the *BbsI*-linearized pSpCas9(BB)-2A-GFP plasmid, as described by Ran et al. 2012. After ligation, bacterial transformation and isolation of plasmid DNA using the Wizard Plus SV Miniprep DNA Purification System (Promega), desired sgRNA inserts were confirmed using Sanger sequencing. To enable excision of the SVA, two recombinant plasmids containing the desired sgRNA inserts to target the 5′ and 3′ end of the SVA were co-transfected into HEK293 cells using TurboFect (Thermo Fisher Scientific). Untransfected unmodified HEK293 were used as a negative control. Cells transfected with non-target gRNA Cas9 constructs (ntgRNAs) were used as an additional control to assess for potential confounding effects of transfecting cells with Cas9 machinery on gene expression. 72 h post transfection, gDNA was purified using the GenElute Mammalian Genomic DNA Miniprep Kit (Sigma). PCR was used to verify the presence or absence of the SVA within the transfected cultures. Cultures showing evidence of an excised SVA were plated at a density of 1000 cells per 10 cm dish, to allow growth of single cells into colonies. When colonies were visible, they were selected and grown in duplicate until 70% confluent. Cell lysates from one duplicate culture were prepared for use as a direct template using DirectPCR Lysis Reagent (Viagen, California, CA, USA) and screened for modifications using PCR genotyping. Primer sequences and thermal cycles are given in Appendix A. Successful excision of the SVA was determined using gel electrophoresis. Complete deletions were sequence verified using the forward PCR primer used in PCR genotyping. Sequencing was conducted externally by Source Bioscience (UK).

### 4.6. qPCR

Total RNA was extracted from HEK293 cell cultures using the Monarch Total RNA Miniprep Kit (NEB) and treated with DNaseI (Thermo Fisher Scientific). cDNA was synthesised using GoScript Reverse Transcriptase Kit (Promega). Quantitative PCR (qPCR) was performed on the Stratagene Mx3005P Real-Time PCR System (Agilent, Crawley, UK) using the GoTaq qPCR Master Mix (Promega). Reactions were set up in triplicate. *ACTB* was used as a reference gene. Target genes were *TRPV1* and *TRPV3*. Relative quantification of target genes was calculated against the reference gene using the delta-delta Ct (2-ΔΔCt) method. Statistical analysis was performed using the 2-sample *t*-test in Minitab version 19.

## Figures and Tables

**Figure 1 ijms-22-01911-f001:**
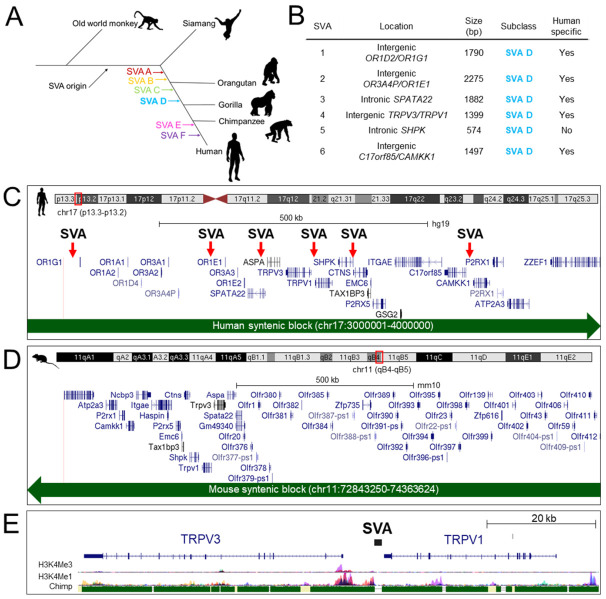
Chr17p13.2 is enriched for human specific SVA D insertions. (**A**) Schematic showing the emergence of short interspersed nuclear element-variable number tandem repeat (SINE-VNTR)-*Alu* (SVA) subclasses throughout primate evolution. SVA subclasses A, B, and C are evolutionarily older and are found in multiple primate species, whereas subclasses SVA E and SVA F emerged following divergence with the chimpanzee last common ancestor and are therefore human specific (adapted from Wang et al. 2005) (**B**) Summary of SVA D insertions at chr17p13.2. (**C**) A gene dense region (chr17:3000001–4000000, hg19) at chr17p13.2 contains 6 SVA D insertions. (**D**) The syntenic region in the mouse genome (chr11:72843250–74363624, mm10) contains orthologous genes conserved at chr11qB4-B5 in the opposite orientation to that displayed in the human genome. (**E**) UCSC image showing human specific SVA insertion at the TRPV1/TRPV3 intergenic region respect to adjacent genes. H3K4Me3 and H3K4Me1 histone marks are shown, highlighting regulatory domains. Conservation with the chimpanzee genome is also provided to assess SVA status as primate or human specific.

**Figure 2 ijms-22-01911-f002:**
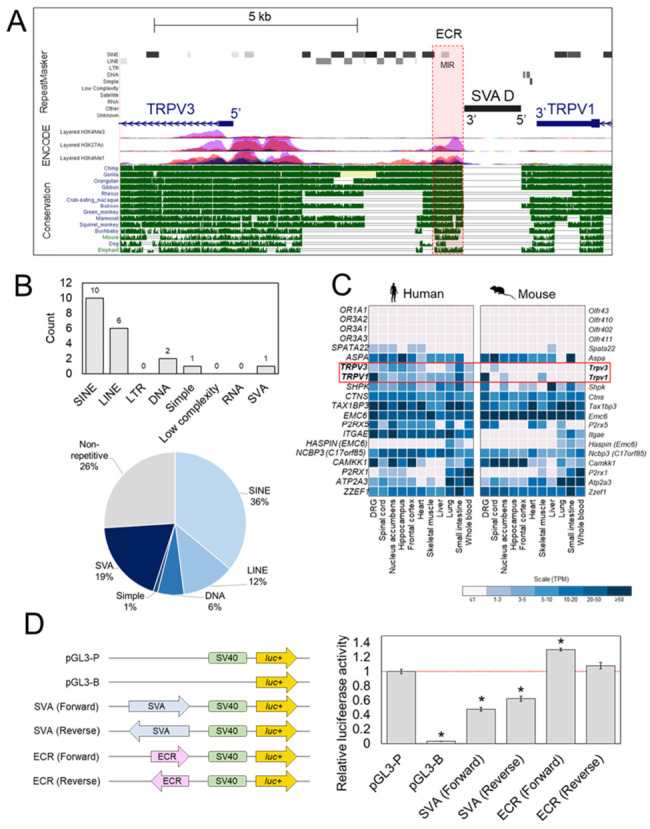
Intergenic region between *TRPV1* and *TRPV3* contains human-specific SVA insertion predicted to function as regulatory domain. (**A**) UCSC image of intergenic region between *TRPV1* and *TRPV3*. RepeatMasker displays repetitive DNA including SVA insertion (ECR containing MIR element is highlighted in red). (**B**) Analysis of TE composition of the intergenic sequence between *TRPV1* and *TRPV3*. (**C**) RNA-seq data showing normalized gene expression values (expressed as transcripts per million; TPM) of genes encoded at chr17p13.2 in human and orthologous genes encoded at chr11qB4-B5 in mouse. (**D**) Relative luciferase activity of pGL3-P reporter gene constructs with SVA (chr17:3466973–3468374) and ECR sequences (chr17:3466258–3466820) cloned upstream of the minimal SV40 promoter (green box) and transfected into HEK293 (*n* = 4). * *p*-value <0.05 (2-sample *t*-test).

**Figure 3 ijms-22-01911-f003:**
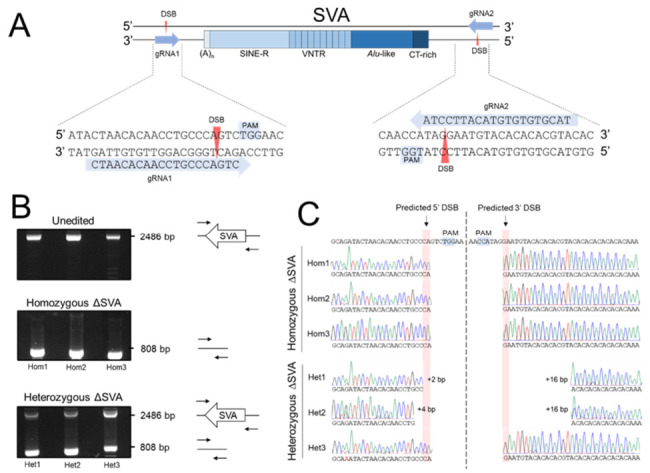
Generation of clonal cell lines containing knockout of entire SVA sequence. (**A**) Schematic of dual-gRNA strategy designed to delete entire SVA sequence. gRNAs were designed to anneal to short 20 bp sequences upstream of protospacer adjacent motif (PAM) sites (5′-NGG-3′) situated on either side of the SVA. Cas9 generates double strand breaks 3–4 bp upstream of the PAM sequence. The SVA is then excised before the two ends are repaired typically by NHEJ, resulting in deletion of the SVA. Schematic not to scale. (**B**) PCR screening showing isolation of several homozygous ΔSVA clones with all SVAs deleted and heterozygous ΔSVA clones demonstrating presence and absence of SVA alleles. (**C**) Sequencing analysis across breakpoints in PCR products.

**Figure 4 ijms-22-01911-f004:**
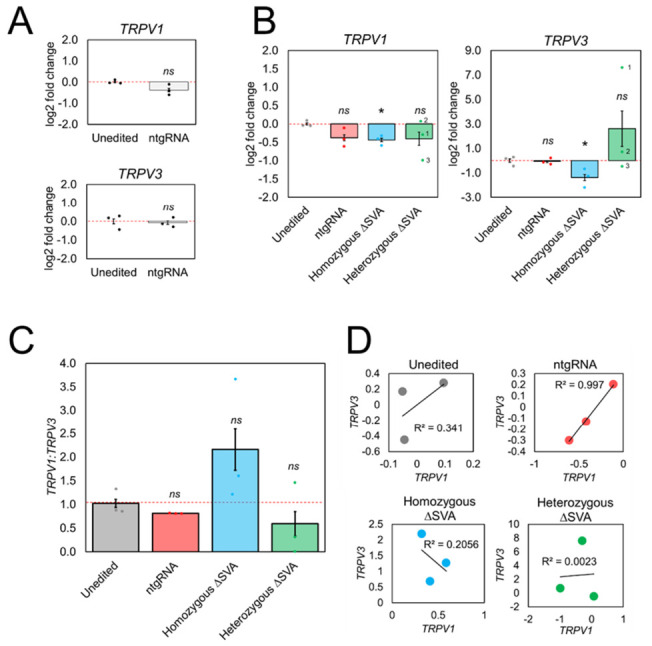
Clustered regularly interspaced short palindromic repeats (CRISPR)-mediated deletion of SVA in HEK293 disrupts co-expression of *TRPV1* and *TRPV3* mRNA. (**A**) *TRPV1* and *TRPV3* mRNA expression in cells transfected with ntgRNAs compared to unedited HEK293. (**B**) mRNA expression in unedited cells, ntgRNA controls, homozygous ΔSVA clones, and heterozygous ΔSVA clones. (**C**) Ratio of *TRPV1*:*TRPV3* mRNA levels. (**D**) Pearson’s correlation of *TRPV1* and *TRPV3* mRNA in all cell lines. Abbreviations; not significant (ns), * *p*-value < 0.05 (2-sample *t*-test).

## Data Availability

Not applicable.

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
