# Peer review of "CRISPR Deletion of a SVA Retrotransposon Demonstrates Function as a *cis*-Regulatory Element at the *TRPV1/TRPV3* Intergenic Region"

_ijms, 2021, doi:10.3390/ijms22041911_

Round 1
Reviewer 1 Report
Dear Authors,
I appreciate reading the manuscript, as it was clear and well written, with good methodology and no over interpretation of the results. I also think that the figures are very clear and appropriate. It was definitely pleasant to review. I have one major concern, and the rest are overall minor points.
Major point:
In the manuscript, as well as in the listed reference of Ray et al (2018), listed as the source of the transcription values in TPM, I could not find (although I admittedly did not do the most extensive research I could have had) if the RNAseq data was done using directional libraries or not. As I work myself on the regulatory impact of transposable elements, I have seen more than often that transposable elements inserted downstream of genes (such as it is the case here for SVA and TRPV1) do increase transcription, but in the form of a stable antisense transcript. Here I am really curious to know what you really quantify in qPCR: a normal TRPV1 transcript, which would strengthen the fact that SVA can act as a "distal" promoter when not inserted upstream of the gene, or a stable AS transcript. Same with the transcription ratios you use in your plots. Do the values for TRPV1 really come from the sense protein coding transcript, or are they a mix of sense and antisense transcripts?
Line 93: For scientists interested in the regulatory impact of transposable elements, but not familiar with the Human model, I believe it would be a good addition to very briefly explain the origin of the HEK293 cell line and why you chose this cell line for your experimentation. My question may sound naive, as I am a plant scientist, but to which extent can the results obtained from a transformed cell line be applied to human in general? Is the expression profile of TRPV3 and TRPV1 in this cell line similar to what is usually found in its tissue of origin?
Figure 1: in my copy of the manuscript, figures 1D to 1F are a bit difficult to read, especially to figure the orientation of the gene. Increasing a little the size of figure 1 would be good.
Line 246 & Figure 4 D: I was really impressed by the 0.997 R2 correlation for the ntgRNA, as I would have expected something similar to the unedited cells. Something is clearly going on here, and it seems too "perfect" to be random. Do you have any explanations here? I think this should be mentioned in the discussion.
Lines 288-292: It is very likely to have been already discussed elsewhere, but I would appreciate that the authors add 1-2 lines with hypotheses to explain why SVA have a repressive impact on reporter models. This is something I found very counter intuitive and it puzzled me during my reading of the results.
Line 403: I think that Ray et al. 2018 should point toward reference #21 and not reference #5, according to your bibliography.
Author Response
Comment 1
In the manuscript, as well as in the listed reference of Ray et al (2018), listed as the source of the transcription values in TPM, I could not find (although I admittedly did not do the most extensive research I could have had) if the RNAseq data was done using directional libraries or not. As I work myself on the regulatory impact of transposable elements, I have seen more than often that transposable elements inserted downstream of genes (such as it is the case here for SVA and TRPV1) do increase transcription, but in the form of a stable antisense transcript. Here I am really curious to know what you really quantify in qPCR: a normal TRPV1 transcript, which would strengthen the fact that SVA can act as a "distal" promoter when not inserted upstream of the gene, or a stable AS transcript. Same with the transcription ratios you use in your plots. Do the values for TRPV1 really come from the sense protein coding transcript, or are they a mix of sense and antisense transcripts?
In the RNA-seq data analysis performed by Ray et al. (PMID: 29561359), libraries were generated using the Truseq RNA sample preparation v2 kit, which is non-stranded, therefore the raw data produced would include protein coding and non-coding transcripts. However, details regarding transcript orientation during bioinformatic analysis was not provided in the Ray et al. manuscript. Regarding the qPCR conducted in our study, we did not perform strand specific reverse transcription when preparing cDNA but the primers for TRPV1 were designed to span exons 1 and 2 (to determine a transcript resulting from the SVA a primer would be required within the SVA sequence). We have supplied additional data in supplementary file 4 (Fig. S2) (referenced on and L180 and L232), including gels and melt curve analysis which shows the specific amplification of a single TRPV1 product. An antisense transcript would not necessarily have the same exons. However, one of the models affecting expression at the region could of course be an antisense mRNA. Furthermore, in such model, it could act specifically on a gene or affect the whole locus by altering chromatin structures (open chromatin).
Comment 2
Line 93: For scientists interested in the regulatory impact of transposable elements, but not familiar with the Human model, I believe it would be a good addition to very briefly explain the origin of the HEK293 cell line and why you chose this cell line for your experimentation. My question may sound naive, as I am a plant scientist, but to which extent can the results obtained from a transformed cell line be applied to human in general? Is the expression profile of TRPV3 and TRPV1 in this cell line similar to what is usually found in its tissue of origin?
HEK293 originate from human embryonic kidney cells. We have added the statement at L180-183: “HEK293 was chosen as a model cell line to conduct CRISPR due to its high transfection efficiency, which was found to be a limiting factor in genome editing efficiency in other cell lines we tested (e.g. HAP1 and SH-SY5Y) in early stages of protocol development (data not shown).” Whilst these cells are suitable to perform in vitro assays, and expression of both TRPV1 and TRPV3 has been documented in human kidney tissues (PMID: 29970964, PMID: 29914124), it is not appropriate to extrapolate the results from HEK293 in this study to human kidney tissues as the molecular phenotype of HEK293 does not recapitulate the tissue specificity of kidney cells (PMID: 26026906). To summarise, this is not possible to determine as the cell line models a window of possible expression patterns. That itself could be modified by cell cycle, passage number and other parameters/exposures whereas in the tissue of origin, the expression would likely be dynamic also.
Comment 3
Figure 1: in my copy of the manuscript, figures 1D to 1F are a bit difficult to read, especially to figure the orientation of the gene. Increasing a little the size of figure 1 would be good.
Reviewer 2 also requested some changes to figure 1 (see review 2, comment 1). The full list of changes made are given below.
The panels in figure 1 have been reordered:
- 1A now shows the schematic of SVA evolution. We also added colours to SVA subclass labels to make it easier for the reader to distinguish between each subclass.
- 1B now shows the table with information for each SVA at chr17p13.2. We continued to colour labelling of SVA D to emphasise to the reader that all SVAs at this location belong to the same subclass.
- 1C now shows the screenshot of genes from UCSC hg10. It has been made larger to make genes easier to read.
- 1D now shows the screenshot of genes from UCSC mm10. It has been made larger to make genes easier to read.
- 1E now shows the TRPV1 and TRPV3 complete coding region, with SVA insertion shown at the intergenic region.
- E-I which show a more in detail view of SVAs in respect to their nearest coding genes have been removed from this main figure and supplied as a new supplementary file 1 (Fig. S1).
Figure legend has been modified accordingly.
Figures within the text have been amended to reflect the order of appearance.
Comment 4
Line 246 & Figure 4 D: I was really impressed by the 0.997 R2 correlation for the ntgRNA, as I would have expected something similar to the unedited cells. Something is clearly going on here, and it seems too "perfect" to be random. Do you have any explanations here? I think this should be mentioned in the discussion.
When looking at the data from qPCR in unedited cultures, all dCt values for TRPV1 (i.e. difference between ACTB and TRPV1) were consistent in unedited HEK293 (10.5-10.6), indicating that TRPV1 levels were consistent in these cultures in respect to the reference gene. For TRPV3, two unedited cultures had the same dCt values for TRPV3 and ACTB (12.3-12.4) whereas one culture showed a slightly larger difference at 13.1 dCt – which ultimately showed a slight decrease in TRPV3 in one culture compared to the other two cultures. The small change in expression in this replicate in respect to the small range of expression values for TRPV3 (<0.7) and TRPV1 (0.14) between each replicate was enough to perturb the correlation. We have added a statement on L260-264 to address this: “It should be noted that a small decrease in TRPV3 expression in one unedited replicate was observed, however TRPV1 expression remain consistent across all unedited replicates. Given the small range in expression values between replicates, this small decrease in TRPV3 was enough to decrease the strength of the positive correlation in unedited replicates.”
Comment 5
Lines 288-292: It is very likely to have been already discussed elsewhere, but I would appreciate that the authors add 1-2 lines with hypotheses to explain why SVA have a repressive impact on reporter models. This is something I found very counter intuitive and it puzzled me during my reading of the results.
Reviewer 2 also raised this point, therefore the answer to review 2, comment
Given our previous published experience with VNTRs, we were also expecting SVAs to be active. However perhaps in human cells they are repressed by the action of ZNF transcription factors. We would not like to speculate in this manuscript, but it is a focus of current research. We do know other SVAs in similar experiments are also repressive in most basal level models. One recent paper described the binding of ZNF91 to the SVA VNTR in a reporter construct as a mechanism of repression in cell lines. A statement, and the reference has been added to the manuscript discussion at LX:
Comment 6
Line 403: I think that Ray et al. 2018 should point toward reference #21 and not reference #5, according to your bibliography.
We have corrected the reference at L403 (now L418) and thoroughly cross referenced the bibliography throughout.
Reviewer 2 Report
In this study Price et al. studied the effect of deleting an SVA retrotransposon on the expression of TRPV1 and TRPV3 in a the HEK293 cell line using the CRISPR-CAS gene editing system. The authors study the effects in both hetero- and homozygote deletion lines and observe significant decrease in both TRPV1 and TRPV3 expression in homozygous deletion cells. These results constitute an important addition to the growing knowledge on the regulatory impact of transposable elements.
The study is overall well performed and presented. My comments relate mostly to clarification of results.
COMMENTS
- L44: Perhaps reorder panels in Figure 1 after the order they appear in the text.
- L67: iPSCs is not previously defined in text.
- Figure 1: The text on axes and the plots themselves are too small for reading in panels D-I. One suggestion is to retain only panel D of these six (D-I), show it larger and present the rest in the supplementary materials.
- L153: I suggest making a new section for the text following L153 until the end of the current section.
- L178: I think that it would be relevant to see the expression level TRPV1 and TRPV3 in HEK293 cells to be able to compare with the data in Figure 2C.
- L219: PAM not previously defined and not an obvious term for the non-gene edit reader.
- L236-L237: The sentence “All gene expression data in edited cell lines was compared against unedited HEK293.” is already mentioned on L227-L228.
- L256: Why do you see such a perfect R2 in the ntgRNA treatment but not for unedited cells?
- L263-L265: Here it could be prudent to acknowledge that you have low power to detect co-expression trends with only 3 replicates.
- Subheadings in the Discussion section would increase readability.
- One thing that is confusing is that you report a repressive effect of SVA in the pGL3-P system but deletion of the SVA in the HEK293 cells lowers the expression compared to unedited cells. Does this mean that the SVA in this context is not repressive? It would be nice to see a discussion on these observations.
Author Response
Comment 1
L44: Perhaps reorder panels in Figure 1 after the order they appear in the text.
Reviewer 1 also requested changes to figure 1 therefore a full list of amendments regarding this figure are provided in response to review 1, comment 3.
Comment 2
L67: iPSCs is not previously defined in text.
We have elaborated in text at L67.
Comment 3
Figure 1: The text on axes and the plots themselves are too small for reading in panels D-I. One suggestion is to retain only panel D of these six (D-I), show it larger and present the rest in the supplementary materials.
Reviewer 1 also requested amendments to Fig 1, therefore a full description of changes made has been given in section: review 1, comment 3.
Comment 4
L153: I suggest making a new section for the text following L153 until the end of the current section.
We created a new section with header “Reporter gene assays support regulatory potential of SVA at TRPV1/TRPV3 locus” at L154.
Comment 5
L178: I think that it would be relevant to see the expression level TRPV1 and TRPV3 in HEK293 cells to be able to compare with the data in Figure 2C.
We did not perform any RNA-seq in HEK293 to compare levels with figure 2C. However, we have provided a gel image with confirmed TRPV1 and TRPV3 mRNA expression in HEK293 (new supplementary file). In addition, we have further provided quality control data showing validation of the qPCR assay in supplementary file 4 (Fig. S2).
Comment 6
L219: PAM not previously defined and not an obvious term for the non-gene edit reader.
We have defined PAM at L219 and included in abbreviations list.
Comment 7
L236-L237: The sentence “All gene expression data in edited cell lines was compared against unedited HEK293.” is already mentioned on L227-L228.
This duplication at L236-L237 has been removed.
Comment 8
L256: Why do you see such a perfect R2 in the ntgRNA treatment but not for unedited cells?
Reviewer 1 also raised the same point, therefore the answer to both is provided in the section: review 1, comment 4.
Comment 9
L263-L265: Here it could be prudent to acknowledge that you have low power to detect co-expression trends with only 3 replicates.
We have added the statement to L272-273: “It should be acknowledged that the statistical power in this study was limited due to the number of biological replicates (n=3).”
Comment 10
Subheadings in the Discussion section would increase readability.
We have added the following subheadings within discussion:
- Human specific SVA insertion at the TRPV1/TRPV3 locus identified as a candidate CRE (L286)
- CRISPR-Cas9 deletion of SVA in HEK293 generated homozygous and heterozygous clones (L308)
- TRPV3 expression was significantly decreased in homozygous ΔSVA clones (L323)
- TRPV1 and TRPV3 mRNA expression was highly variable in heterozygous ΔSVA clones (L347)
- CRISPR deletion of SVA at TRPV1/TRPV3 locus demonstrates in situ function as newly evolved CRE (L376)
Comment 11
One thing that is confusing is that you report a repressive effect of SVA in the pGL3-P system but deletion of the SVA in the HEK293 cells lowers the expression compared to unedited cells. Does this mean that the SVA in this context is not repressive? It would be nice to see a discussion on these observations.
We can’t easily compare exogenous and endogenous function of the regulatory domain due to a) chromatin structures at the endogenous gene not present on the transient expression construct b) the context of adjacent regulatory domain in CRISPR experiments and c) modulation of chromatin structure at the locus by deletion. There could be multiple repressors all responsive to distinct pathways and they may not be acting synergistically or additively. As outlined above this model is one window on the possible dynamic expression at the TRPV1/3 locus and it could be very different in response to distinct challenges. We have added the text at L298-303: “Furthermore, a study of repression in exogenous SVA reporter constructs showed repressive binding of the HUSH complex to the central VNTR, which induced methylation at the SV40 promoter. Overall, trends from various reporter models show a repressive effect of SVAs, however this does not necessarily reflect the endogenous role of SVAs in situ, as they are simplistic and cannot account for additional factors like chromatin structure or adjacent regulatory sequences.”